# Experiences of Adults High in the Personality Trait Sensory Processing Sensitivity: A Qualitative Study

**DOI:** 10.3390/jcm10214912

**Published:** 2021-10-24

**Authors:** Sharell Bas, Mariëtte Kaandorp, Zoë P. M. de Kleijn, Wendeline J. E. Braaksma, Anouke W. E. A. Bakx, Corina U. Greven

**Affiliations:** 1Radboud University Medical Center, Donders Institute for Brain, Cognition and Behaviour, Department of Cognitive Neuroscience, 6525 EN Nijmegen, The Netherlands; corina.greven@donders.ru.nl; 2Rotterdam School of Management, Erasmus University, 3062 PA Rotterdam, The Netherlands; mkaandorp@rsm.nl (M.K.); zdekleijn@hotmail.com (Z.P.M.d.K.); 3Behavioural Science Institute, Radboud University, 6500 HE Nijmegen, The Netherlands; wjebraaksma@gmail.com (W.J.E.B.); a.bakx@fontys.nl (A.W.E.A.B.); 4Fontys University Child and Education, Fontys University of Applied Sciences, 5022 DM Tilburg, The Netherlands; 5Karakter Child and Adolescent Psychiatry University Center, 6525 GC Nijmegen, The Netherlands; 6King’s College London, Institute of Psychiatry, Psychology and Neuroscience, Social, Genetic and Developmental Psychiatry Centre, London SE5 8AF, UK

**Keywords:** sensory processing sensitivity, highly sensitive person, qualitative research, adults, coping

## Abstract

Sensory processing sensitivity (SPS) is a personality trait reflecting inter-individual differences in sensitivity to negative and positive environmental information. Being high in SPS is associated with increased stress-related problems if environments are unfavourable but also appears to enhance one’s ability to benefit from health-promoting environments. In understanding SPS, therefore, lies the potential for innovating the ways we use to promote mental health. However, as a young research field, the core characteristics of SPS are yet debated. Qualitative research interviewing highly sensitive adults is important to conduct ecologically valid research connected with the complex realities of people. This study was the first to systematically report the perceptions and experiences of SPS characteristics in adults high in this trait. Semi-structured interviews (*n* = 26) were analysed thematically and described following consolidated criteria for reporting qualitative research. Six themes emerged: (1) emotional responding; (2) relatedness to others; (3) thinking; (4) overstimulation; (5) perceiving details; and (6) global SPS characteristics. With regards to coping with negative consequences of high SPS, the main themes were: (1) reducing sensory input and (2) psychological strategies. We gained fine-grained information on experiences of adults high in SPS and derived new hypotheses regarding the fostering of well-being related to high SPS.

## 1. Introduction

Everybody is, for survival, sensitive to environmental stimuli, although the degree of sensitivity differs between individuals. Sensory processing sensitivity (SPS) captures a continuum of inter-individual differences in sensitivity to environmental information in an evolutionarily conserved personality trait, observed in >100 animal species [1]. SPS is heritable (around 45%), and common, with around 20%–30% of the population considered high in SPS (also “highly sensitive”) [2,3,4]. Although SPS correlates with other personality traits, especially negative emotionality and openness to experience, SPS is largely distinct from Big 5 personality dimensions or other temperament traits and may be considered a profile of lower-level personality facets [3,5,6,7,8]. In the past, sensitivity to environments was exclusively regarded as a risk factor. Recent findings suggest that individuals are differentially susceptible to both aversive and supportive environments [9]. As such, high SPS is related to stress-related problems and being more susceptible to overstimulation by sensory input and unfavourable environments, but is also associated with being more susceptible to positive mood induction and benefitting more from psychological interventions [10]. A recently published systematic review showed that individuals high in SPS experience a lower quality of life in physical, mental, emotional, and social areas and also highlighted some aspects of SPS that were associated with positive outcomes [11]. In understanding SPS, therefore, lies the significant potential for innovating the ways we promote (mental) health of individuals high in SPS. However, despite the by-now recognised scientific relevance of SPS, its scientific base is still recent. In the present qualitative study, we shed further light on positive and negative aspects of high SPS, as experienced by adults high in this trait, with the ultimate goal to contribute to optimising phenotyping and validation of the SPS trait.

The SPS construct was first described in 1997, based on a literature review in combination with qualitative interviews [3]. Current theorising suggests that SPS is a unitary construct that comprises of the following key characteristics: greater depth of processing, emotional reactivity and empathy, sensitivity to subtleties, and overstimulation [1,12]. To assess SPS, the Highly Sensitive Person (HSP) Scale was developed. The scale was derived from the qualitative interviews in the 1997 study, and most quantitative research on SPS is based on this questionnaire. Although designed to capture a unitary construct, factor analyses revealed that the HSP scale has three subscales: ease of excitation (EOE), low sensory threshold (LST), and aesthetic sensitivity (AES) [8]. The first two subscales correlate with negative outcomes such as increased stress, depression, and anxiety, and the latter subscale with positive outcomes such as enhanced response to psychological intervention and positive mood induction [10]. The HSP scale items were later repeatedly shown to load onto an overarching “sensitivity”-factor that reflects variance common to all HSP scale items in addition to three orthogonal factors that capture variance specific to EOE, LST, and AES items [4,7,13]. This may reconcile the seemingly contradictory views of a unitary construct and different subscales.

Yet, there are still limitations to both the HSP scale and its theorising. Interpretation of questionnaire results is always limited by the content validity of the scale. The results from the HSP questionnaire differ from current theorising in that there is an overemphasis on overstimulation in the HSP scale, whereas depth of processing is only indirectly captured in the aesthetic sensitivity component [10]. Furthermore, the few items in the HSP scale referring to positive aspects of SPS (AES) appear to be measured less reliably [13]. Lastly, although SPS theory has been updated based on empirical evidence, certain hypothesised central characteristics are not theoretically fleshed out fully yet. To provide one example, empathy consists of several characteristics such as affective and cognitive empathy [14] that involve distinct neural pathways [15]; yet the role of empathy in SPS remains unclear, as well as which aspects are relevant in SPS. It, therefore, remains debated which key characteristics are central and essential to SPS.

Given the limitations of the main scale to assess SPS, a qualitative approach that gains independence from the scale is highly suited and complementary to quantitative research. Since SPS is a young research field, qualitative research on the everyday experiences of highly sensitive individuals themselves could lead to new hypotheses for ecologically valid quantitative studies that are connected to the complex realities of people and are better able to capture heterogeneity in experiences [16,17]. Methodological quality standards concerning transparency and systematicity have led to qualitative research gaining influence in research [18]. Further, qualitative research fits international research agendas to do more justice to the perspectives of people who the research is about, especially for a topic such as SPS, which is receiving increased societal attention. However, the qualitative interviews on which the HSP scale and its adaptations are based were not systematically reported [3]. The only systematically reported qualitative study on SPS so far [19] focused on enablers and barriers of well-being in highly sensitive adults.

To conclude, substantial progress in phenotyping and validation of SPS has been made, but more insights are needed on the everyday experiences of highly sensitive individuals themselves. Therefore, a qualitative inductive approach was chosen to explore SPS, using thematic analysis and following consolidated criteria for reporting qualitative research (COREQ) criteria [20]. This is the first systematically reported qualitative study that focuses on perceptions and experiences of SPS characteristics by high-SPS individuals. The overall aim was to study what high-SPS adults consider characteristics of SPS and their experience of these characteristics as positive or negative. As secondary aims, we wanted to find out how adults high in SPS cope with negative consequences of the trait and what they need for their well-being and to explore the effects of finding out they were highly sensitive.

## 2. Materials and Methods

### 2.1. Recruitment and Sampling

This study included adults aged 25 to 50 years who considered themselves highly sensitive and were fluent in Dutch. The age range was chosen to balance the heterogeneity and homogeneity of the sample. The study was advertised, primarily, by two Dutch SPS knowledge centres and a charity with a large and active database of people high in SPS, and secondarily on social media pages of the researchers. The advertisement provided brief information about SPS and asked for participation in a study on experiences of highly sensitive adults. Participants had to register their interest to be interviewed via the website of the Donders Institute, where they completed a short form about their demographics, contact details, and other background questions.

In total, 494 people signed up between 3 February and 16 November 2020 (Figure 1). To ensure the representation of various subgroups, participants from eight subgroups were selected by stratified random sampling. The subgroups were created based on three variables: age (25–37 and 38–50 years), sex, and highest level of education (Appendix A). The registration form on the Donders Institute website remained live until data saturation was reached [21]. A total of 35 people was invited for the interviews, of whom 26 participated.

We followed the consolidated criteria for reporting qualitative research (COREQ, Appendix A) [20]. All participants provided informed consent for audio-recording the interviews. The Medical Ethics Review Committee of Radboudumc confirmed that the Medical Research Involving Human Subjects Act does not apply to this study, and thus no official approval of the committee was required (2019.6021). The privacy was observed in accordance with the principles of the Declaration of Helsinki. Participants received a €15 gift card as an incentive for their participation.

### 2.2. Questionnaires

The demographic variables sex, age, and highest level of education were assessed for selection of participants and demographics. Education level was dichotomised, and a higher education level was defined as having completed a university or university of applied sciences education [22].

To check high SPS, we used the established 27-item Highly Sensitive Person Scale (Cronbach’s α = 0.81) [3]. Example items of the HSP scale are: “Do you find it unpleasant to have a lot going on at once?” (ease of excitation subscale; 12 items, Cronbach’s α = 0.71), “Are you bothered by intense stimuli, like loud noises or chaotic scenes?” (low sensory threshold subscale; 6 items, Cronbach’s α = 0.51), and “Are you deeply moved by the arts or music?” (aesthetic sensitivity subscale; 7 items, Cronbach’s α = 0.80). Exploratorily, we added the new 30-item Sensory Processing Sensitivity Questionnaire (Cronbach’s α = 0.88) (De Gucht, unpublished). The SPS scale is a new scale that is currently under review. All items were scored on a 7-point Likert scale ranging from “not at all” to “completely”. The two questionnaires correlated highly, r = 0.851, *p* < 0.001.

To further aid the phenotyping of participants, Big 5 personality traits were assessed by a validated Dutch adaptation of the 60-item Big Five Inventory 2 (BFI-2) [23,24]. This questionnaire measures extraversion, agreeableness, conscientiousness, negative emotionality, and openness to experience. The items were scored on a 5-point Likert scale ranging from “strongly disagree” to “strongly agree”.

### 2.3. Interview Protocol

We conducted semi-structured interviews using a topic guide with open-ended questions about what the participants considered to be key characteristics of SPS and how these characteristics are perceived and experienced (e.g., positive or negative). Moreover, we asked questions about coping with the negative sides of being highly sensitive and what they need for their well-being and the moment and impact of realising being highly sensitive. Furthermore, questions were asked about well-being and psychological and somatic problems. We added a question to check whether the corona situation influenced the answers provided in the interview, and all participants said it did not.

### 2.4. Conducting the Interviews

Two pilot interviews, not included in the analysis, were conducted with self-identified, highly sensitive people from our own network. Minor changes were made in the interview protocol based on these interviews. The selected individuals were invited by e-mail or phone call. Reminder emails were sent when necessary. Before the interview, phone contact occurred with almost all participants (*n* = 24) to schedule the interview and establish a relationship. The interviews were conducted by the authors of this study, who are four experienced and certified female interviewers (MSc and PhD) and two trained female research interns (MSc level). The participants did not know any further details about the researchers’ goals and interests. Because of the COVID-19 pandemic measures, only four interviews occurred in person at a research institute. The other interviews were conducted by video call (*n* = 21) and phone (*n* = 1). The interviews were in Dutch and lasted between 52 and 110 min, with an average of 75 min. Conducting interviews was performed in parallel with analysing the data between February and December 2020.

The interviews were transcribed verbatim, omitting information that could be traced back to the participant. The transcripts were sent to the interviewees for comment and correction. Five interviewees noted minor errors, and six interviewees provided minor additions. Furthermore, notes were made during and directly after the interview to develop ideas about the study. We also invited all interviewed participants to a digital meeting in April 2021, where we presented our preliminary results and gathered feedback from the participants.

### 2.5. Data Analysis

The data were analysed thematically in Atlas.ti 8.4.20 using a constant comparison method [25,26,27]. We analysed the data inductively by first creating codes that were close to the data (i.e., open coding). A coding scheme was created that developed during the process. Eight interviews were coded by multiple coders independently and were discussed. Regular team meetings were held in order to review new insights from the interviews and to (re)define the codes. When consensus was reached, all interviews were coded by two researchers. The first coder created the codes, and the second coder checked whether the codes were assigned according to the latest coding scheme. The first author coded all interviews as a first or second coder. The codes were categorised into (sub)themes using a constant comparison method within and between interview transcripts (i.e., axial and selective coding) [25,26,27]. Some codes fit into multiple themes and were categorised accordingly. The (sub)themes, as well as data saturation, were discussed in several team meetings. In the Results section, only subthemes are presented that were mentioned by at least ten participants. Subthemes mentioned by four to ten participants are presented in the “Other” paragraphs.

## 3. Results

### 3.1. Participants

The average age of the sample (*n* = 26; 13 males, 14 lower educated) was 37.41 years (SD = 6.73).

The sample scored high on the SPS scales (Table 1; see Appendix A for mean scores on HSP subscales); 24 participants (92%) scored above the cut-off for a high-sensitive group on the total score of the HSP scale [4]. We did not exclude the two participants who scored within the medium-sensitivity range because the cut-off value is not validated yet in a Dutch sample, and these two participants provided answers that were similar to the answers of the other participants. Comparing the scores on the Big 5 dimensions with a representative Dutch sample [24], our sample appeared to score higher on openness to experience and negative emotionality and had similar scores on the other three Big 5 dimensions. These findings correspond with a recent meta-analysis on personality correlates of SPS [5].

### 3.2. Characteristics of SPS: Perceptions and Experiences

A total of 6 themes and 20 subthemes were identified, reflecting the participants’ perceptions and experiences of SPS characteristics (Figure 2). The six themes are: (1) emotional responding; (2) relatedness to others; (3) thinking; (4) overstimulation; (5) perceiving details; and (6) global SPS characteristics. The (sub)themes are presented in descending order of the number of participants who mentioned them. The (additional) example quotes, as well as the participant characteristics for each quote, are provided in Appendix A.

#### 3.2.1. Emotional Responding (*n* = 26)

The first of three subthemes, the experience of negative emotions, included a strong response to negative events, to emotions or behaviour of others (e.g., anger and rejection), and to negative media (e.g., news and violence). For example, one participant stated that negative comments hurt her more deeply than they would hurt other people:

P20: “When I am at a party for example and I say, like: ‘What nice tableware’ or something like that and somebody says, like: ‘Well, I think it actually looks terrible’, then I can suddenly start thinking: ouch! That it cuts me deeper than I perceive it cutting people around me. And then I really feel like a scared little animal that’s easily hurt or something.”

The second subtheme, the experience of strong positive emotions, included being touched by music and other forms of arts and the ability to intensely enjoy the small things in life:

P9: “Very small things, I can intensely enjoy those, yeah. Small things that people often overlook. A breeze through your hair for example, or the sun that shines, or leaves that make that rustling sound when you walk over them in autumn.”

As a third subtheme, participants indicated needing more time to process emotions, particularly negative emotions. One participant reflected on his response to watching violent videos:

P22: “It seems that many people don’t mind it if they, for example, look at a video which contains violence. They can just scroll away. But with me it actually sticks. It hits me really hard and it just sticks.”

#### 3.2.2. Relatedness to Others (*n* = 25)

Five subthemes emerged. First, participants reported they readily notice or feel other people’s emotional states and the atmosphere of a situation. For example, they can sense when people had an unresolved disagreement:

P10: “I can just look around the group and immediately spot who is feeling well and who is fighting as a couple, or where there’s tension.”

Second, attention to the needs of others was perceived as a core feature of SPS, i.e., finding it important to contribute to other people’s well-being (“pleasing”), sometimes at the cost of their own needs. For example:

P8: “When I work together with colleagues for example, I very quickly know like: this colleague likes this, this colleague likes that. So, I often get told that collaborating is nice, because I, of course, think like: oh, that one likes to do things that way, and that one likes to do things that way. But more often than not I forget to think about my own interests.”

Third, the ability to understand other people’s emotions and intentions was mentioned. For example, participants reported being able to take the perspective of others:

P14: “When I first make contact with children, especially when they are new arrivals, I very quickly notice […] I start reacting very enthusiastically or I need to really stay calm. Yeah, for the most part I quickly realise how a child is put together and how I can respond to the situation.”

Fourth, participants described situations in which they acted on what they knew about other people’s emotional or cognitive states. For example, when they noticed someone did not feel well, they would approach this person to help:

P21: “When someone says, for example, like ‘sure, I’m okay’, I might want to ask like ‘hey, how are you really? Because I see this and this.’”

Fifth, participants expressed feeling a deep connection with other people, including family, friends, and people they do not know. They actively look for ways to create these feelings of connection and want to share these feelings with others. Some participants mentioned a downside of this characteristic:

P19: “When I am really good friends with someone, that can be a real deep feeling you know? A true connection, as if it’s family I guess. So, when something negative happens or there’s a fight one time or something, then it kinda really hits me hard.”

#### 3.2.3. Thinking (*n* = 25)

SPS characteristics related to thinking were categorised into three subthemes. First, participants indicated they worried and ruminated more than others about both private and societal events. This worry includes a tendency to relate (negative) things that are being said to themselves. One participant mentioned that rumination prevents him from falling asleep:

P14: “Like when I finally lie in bed and go to sleep, everything keeps like rolling around in my head and I keep mulling things over. And yes, I do mean every time, I can never get to sleep in one go. For example, my ex, they always said good night and they were gone within ten seconds, but I would keep thinking for an hour before I fell asleep.”

Second, participants mentioned thinking and reflecting a lot. For example, they tended to need more time to make decisions:

P6: “Sometimes it takes awfully long for me to make a decision, because I want to weigh all the options and the pros and cons and what everyone’s opinion might be. These are mostly, for example, things we are going to do, and my girlfriend then thinks: get on with it already.”

Third, a higher need for depth and meaning was expressed. For example, in conversations, they prefer exploring topics in-depth over small talk:

P11: “My father always said: ‘you always get so philosophical right off the bat’. And then I think: What are you talking about? I am just talking about the state of my life or how I experience something or think about it and he immediately thinks it’s deep?”

#### 3.2.4. Ease of Overstimulation (*n* = 24)

Four subthemes emerged that are related to ease of overstimulation. The first two subthemes were related to the source of overstimulation. Participants indicated being easily overstimulated, in particular by sensory stimuli, such as noise, lights, scents, and tactile stimulation (clothing). One participant provided an example of overstimulation by visual stimuli at work:

P19: “Well, at my work at [company] we naturally have rather bright lights to nicely illuminate all those [products], but sometimes I would just stand still for a moment […] but then if one of those bright lights might shine on my face in that particular spot, I would step aside, because I know it might cut really deep and would need to be processed, so to speak.”

Overstimulation can also be caused by social stimuli, such as large crowds:

P13: “I actually don’t go to large events in large venues anymore. I do that very rarely because I know I can actually only take it for like an hour or something.”

The third and fourth subthemes concerned the effects of overstimulation. Overstimulation could lead to reduced cognitive abilities, such as being distracted, feeling restless, and not having a clear mind. For example, overstimulation could negatively impact decision-making abilities:

P5: “I actually have a lot of problems making decisions when I experience some overstimulation. Well, I have that after a day of work when I stand in the supermarket, where I think: I can’t take this, I’ll just make spaghetti bolognaise again or whatever.”

Overstimulation can also lead to negatively affected mood, as expressed by the following participant who describes being irritated and angry as a result of overstimulation:

P16: “People who are constantly clicking their pens, oh, please stop. I can’t stand it. I get incredibly annoyed. I don’t know what it does to me specifically. I’m not that far yet. But I get incredibly annoyed by it. I just go off like a firework, I just get angry and rebellious and then I’m like: cut that out.”

#### 3.2.5. Perceiving Details (*n* = 21)

Two subthemes were identified. First, many participants indicated they perceived a greater quantity of information. This was often referred to as a “lack of a filter”:

P24: “I sometimes specifically notice that when you walk past some place while you’re with someone, you might see something really beautiful and the person you’re with just completely misses it or walks past it.”

The second subtheme is a perception in greater detail, by one participant described as perceiving the world in “high definition”. One example is a more detailed perception of visual and auditory cues in non-verbal communication, such as facial expression, body language, or tone of voice.

P9: “You notice a lot more details of things. For example, my wife showed me a picture this morning, a baby picture, never seen it before. ‘Who is that?’ she asks. I say: ‘that’s your father’. But I’ve only known her father since he was a grownup. She asked: ‘I hadn’t even recognised him. How did you know?’ ‘Well, his eyes, his ears, I guess... I dunno.’”

#### 3.2.6. Global SPS Characteristics (*n* = 26)

This theme refers to overarching SPS characteristics that were related to and sometimes described as the consequence of more than one of the abovementioned themes. Three subthemes on global SPS characteristics were defined. The first subtheme concerned self and identity and includes feeling different and having low self-esteem because of their high sensitivity. Moreover, some participants considered being highly sensitive an essential part of themselves:

P4: “It [HSP] really encompasses your entire being.”

As a second subtheme, participants indicated feeling easily stressed and finding it difficult to relax:

P7: “I would like to be able to relax more, that is, in the evenings. I am constantly in ‘hurry-up’ mode.”

Lastly, feeling tired was mentioned as a general trait of SPS, sometimes described as having a lower energy level. For example:

P25: “It’s as if I’m more conscious of those details, those expressions, of what comes back to me. And it’s a lot of fun and actually quite useful. But it can also be quite tiring without you noticing it.”

#### 3.2.7. Other SPS Characteristics

Other (sub)themes include difficulties with (pressure from) work or study, creativity and a greater quantity of ideas, both introversion and extraversion, working hard and perfectionism, intense experiences and a rich inner world, spirituality, associative thinking, a specific physical sensitivity (e.g., caffeine), impulsivity, autonomy, and being able to integrate and draw conclusions from complex, subtle details. Examples of the last subtheme include an increased ability to sense and anticipate problems and to know what to do to prevent or resolve them. With regards to introversion and extraversion, most participants described themselves as an introvert and seeking quiet environments because of their ease of overstimulation, but at the same time, most also described themselves as sociable, and some described the need to external stimulation and being sensation-seeking, suggesting their introversion-extraversion level may depend on the situation and their level of overstimulation.

### 3.3. Strategies and Activities to Improve Well-Being

Two themes, reducing sensory input and psychological strategies, with six subthemes were identified, reflecting the strategies and activities that improve well-being. The (sub)themes are presented in descending order of the number of participants who mentioned them. Example quotes are provided in Appendix A.

#### 3.3.1. Reducing Sensory Input (*n* = 26)

Two subthemes emerged related to reducing sensory input. First, almost all participants mentioned a need for solitude and quiet environments as a strategy to deal with or prevent overstimulation. They expressed a need for time for themselves to relax and recharge. They fill this time with different activities, such as walking in nature, quiet activities, sleeping, and simply “doing nothing”. Second, strategies were mentioned to reduce specific sensory input. For example, some participants reported using noise-cancelling headphones or sunglasses.

#### 3.3.2. Psychological Strategies (*n* = 25)

Three subthemes are related to psychological strategies. First, support from others was important to some participants. For example, they expressed the need to share their experiences with other (highly sensitive) people and to feel understood. Second, several participants used techniques from mindfulness, meditation, and yoga to process information and to let go of negative thoughts. Third, positivity, acceptance, and reflection help deal with negative thoughts and can stimulate or enhance positive thoughts, according to the participants.

#### 3.3.3. Other Strategies

Other (sub)themes include an environment in which they are able to act autonomously, a career that fits with being high in SPS, exercising and a healthy diet, routine and structure, and avoiding watching the news or violence. Moreover, some participants mentioned they are not yet sure what works best for them and try out different strategies.

### 3.4. Learning about SPS and Attitude towards SPS

Participants were on average 30 years old when they found out they were high in SPS, which was on average seven years ago. About half of the participants indicated that going through difficult times, such as psychological or relationship problems, was the indirect cause of learning about SPS. When asked about the direct cause, about half of the participants mentioned that someone else, such as a friend, family member, or psychologist, pointed out to them that they might be highly sensitive. Many participants viewed SPS as an explanation for some of their feelings and behaviour. For example, it felt “like the puzzle pieces fell into place” and “a hundred things suddenly make sense”. Moreover, some participants experienced a feeling of recognition. They realised they were “not crazy” and felt relieved there was “nothing wrong” with them. In addition to that, learning about SPS helped cope with the negative sides of SPS because SPS could serve as a starting point in the search for information.

Most of the participants expressed a high current level of well-being. Moreover, around half of the participants reported internalising problems in the past or in the present, such as burnout, depression, or anxiety, and related these problems to the experience of stress that comes with being highly sensitive. Half of the participants reported experiencing a higher level of well-being after the realisation of being high in SPS, and the other participants did not report a change in well-being. Regarding attitude towards SPS, most participants indicated having a positive attitude, and some expressed a neutral or balanced view towards SPS. Two participants mentioned viewing SPS as entirely negative because of their depression or anger, and they related these symptoms to their high sensitivity. Most of the SPS characteristics were viewed as both negative and positive or as entirely positive. For example, participants appreciated their tendency to help others but also saw the trade-off with regards to their own needs being sacrificed. Overstimulation, tiredness, and stress were exclusively viewed as negative.

### 3.5. Meeting with Participants

About half of the participants (*n* = 14; 8 males, 5 lower educated, age: M = 38.64 years, SD = 7.12) attended the interactive meeting to learn about the preliminary results of this study and provided feedback. The participants noted that the results supported their view on SPS. They also asked questions, e.g., about the differences between SPS and autism spectrum disorder (ASD), ADHD, and giftedness. We took their questions and suggestions into account in the interpretation of the findings.

## 4. Discussion

### 4.1. Summary

The aim of the current study was to contribute to the scientific knowledge on perceptions and experiences of SPS characteristics by high-SPS individuals. Six themes emerged, most of which matched with the hypothesised key characteristics described in the introduction. We also uncovered novel insight into how these themes are experienced, as well as novel subthemes on perceived SPS characteristics. The six themes were mentioned by almost all participants, suggesting SPS entails a specific combination of characteristics. We also found heterogeneity; a number of subthemes came up in only some of the interviews. As secondary aims, we wanted to look at coping with the negative sides of SPS and increasing well-being and the effects of finding out about SPS. With regard to coping, the adults in our study mentioned reducing sensory input and several psychological strategies. Most people expressed a positive attitude towards SPS and reported positive effects of finding out they were highly sensitive.

### 4.2. Phenotyping and Validation of SPS

We identified six themes on perceived key SPS characteristics. Five of these themes fit with the theorised key SPS characteristics [1,12]. Our themes of emotional responding and relatedness to others fit with theorised “emotional reactivity and empathy”, thinking with “depth of processing”, perceiving details with “sensitivity to subtleties”, and overstimulation directly mapped onto “overstimulation”. Moreover, we obtained novel and fine-tuned insight into how each of these themes is experienced.

For example, the role of empathy in SPS is still debated theoretically but came out as a strong theme in the experiences of the participants in this study. Based on the interviews, new insight was won that several aspects of empathy may be increased in high SPS, including affective empathy, cognitive empathy, emotional contagion, and prosocial behaviour [14]. The experience of the empathy-related aspects of attention to others and feeling connected with others was also new in the scientific literature. Worrying and ruminating was another novel perceived characteristic (in the thinking theme) and includes relating information from the outside world to themselves. This may reflect enhanced self-referential processing, which may be a mechanism contributing to increased risk for stress-related problems in high-SPS individuals.

We also identified new (sub)themes not described previously. A sixth theme emerged on perceived global SPS characteristics that concerned self and identity (e.g., feeling different), feeling easily stressed, and feeling tired. Whether these identified experiences are central and essential to high SPS cannot be disentangled in our design, but our results suggest that these experiences may be central to how the trait is experienced by highly sensitive adults, although several participants also reported experiencing the characteristics in the six theme as a consequence of several of the other five themes.

Our sample’s score on the Big 5 questionnaire and their responses on the interviews were in line with quantitative personality research that showed that SPS correlates with negative emotionality and openness to experience [5]. However, our identified themes suggest that it is too simplistic to state that SPS is just a combination of negative emotionality and openness to experience; instead, SPS seems to encompass a specific profile, supporting previous quantitative work on personality profiles of SPS [6].

### 4.3. SPS and Well-Being

Although some of our participants (had) experienced burnout, depressive, or anxious symptoms, they also expressed experiencing a high current level of well-being. This fits well into the two continua model, which describes that measures of (positive) mental health and measures of mental illness are two distinct but correlated continua [28]. Another possible explanation is that individuals who had some mental problems in the past learned to cope with SPS and improve their well-being after learning about SPS.

The strategies and activities of individuals high in SPS found in the current study are in line with a previous study that identified several enablers of well-being for people high in SPS, including mindfulness training and meditation, positive relationships, solitude, and practicing emotional self-regulation [19]. Previous research supports the finding that emotional self-regulation, through positivity, acceptance, and reflection, could be more effective for high-SPS individuals. For example, one study revealed that individuals who were high in connectedness, a characteristic related to SPS in our study, benefit more from a self-compassion exercise to improve happiness and decrease depressive symptoms [29].

Given that SPS is a trait emerging early in life [7,13,30,31], the participants found out relatively late that they were high in SPS. Some participants expressed that if they had known earlier, some of their stress-related problems might have been prevented. As SPS is known to be linked to both stress-related problems but also responding more to positive intervention [10], early awareness, for example by awareness training and societal attention to SPS, could be of help with preventing the personal and societal burden of stress-related problems, as well as the preservation of human capital.

### 4.4. Strengths and Limitations

One strength of the current study is the demographically relatively balanced sample with regard to age, sex, and education level. This increases the ability to capture heterogeneity and is complementary to the group-based analyses in previous quantitative research on SPS. Another strength of the study is that SPS and personality scores supported that the participants were highly sensitive in the way described in the scientific literature.

This study also has other methodological strengths that improved the interpretation of the findings, including following the COREQ criteria [20] and the rigour of the analytic process. For example, the participants were sent the interview transcript for review, and all interviews were coded by two coders. The qualitative approach allowed us to be independent of the content validity limitations of the HSP scale and investigate what self-identified, highly sensitive adults think is central to high SPS.

One limitation of this study is a potential self-selection bias [32]. As the participants were already interested in SPS, their answers could have reflected their knowledge of SPS. However, the rich examples of the perceived key characteristics support the view that participants answered the questions based on their own experiences, and most participants were not familiar with the SPS (scientific) literature. Further, while lay theoretical perspectives can draw on verification over falsification, confuse cause and consequence, or build on misinformation, they can also overlay with scientific theories or influence one another, and produce insight beyond common sense or be counterintuitive [33,34].

Lastly, many of our findings seemed specific to SPS and may be regarded as specific in that the constellation of characteristics (themes) co-occurred in most of our interviewed participants rather than reflecting general human experiences. For example, many participants spontaneously compared their experiences to those of the people around them.

### 4.5. Future Directions

Future research on this topic could study SPS in other demographic groups. First, our findings were based on a Dutch sample, and a previous qualitative study used a U.S. sample [3]. Cultural aspects may influence the expression and experience of SPS, and future studies could therefore study SPS in other countries or cultures. Second, since SPS is a heritable trait and is expressed from early childhood [2,7,13,30,31], knowledge on developmental aspects of SPS could improve early awareness of SPS. Furthermore, future research could also involve triangulation (perspective of others).

Our results give rise to novel hypotheses for follow-up in quantitative research. For example, we hypothesise SPS to be related to enhanced affective and cognitive empathy [14], which can be tested through neurocognitive tasks. To provide another example, future eye-tracking studies could investigate the perception of facial micro-expressions in individuals high in SPS, based on the participants’ reports of having a greater ability to perceive subtle changes in facial expressions that emerged from our study.

Results from our study may be used to optimise the HSP scale by developing questionnaire items people can relate to. The current HSP scale disproportionally captures negative aspects of SPS related to overstimulation and emotional reactivity, with very limited items on cognitive aspects of SPS, empathy, or positive emotional responsivity [3], although theorised to be central to the construct [1,10,12].

The similarities and differences between SPS, ASD, ADHD, and giftedness, which our participants enquired about, could be studied in future research. Sensory sensitivity has been found in both ASD and ADHD [35,36]. However, one key difference between SPS and ASD might be the theory of mind (i.e., cognitive empathy), which high-SPS adults may be expected to score high on [37,38]. With regards to ADHD, one main difference between SPS and ADHD could be the degree of impulsivity, where high-SPS individuals are expected to be more inhibited. Comparing SPS and giftedness, some aspects may overlap [39], but the exact relation between SPS and giftedness is unclear yet. More research is needed to identify the similarities and differences between high SPS, neurodevelopmental disorders, and giftedness.

### 4.6. Practical Implications

During the past decades, the topic of SPS has gained increased interest in society. This interest was also expressed by the high numbers of people who showed interest in participating in our study (almost 500). In the interviews and the meeting with participants, it was expressed that knowing about SPS could already improve well-being. The participants emphasised the importance of spreading knowledge on both the positive and the negative sides of SPS to society, for example, to schools and employers, to reduce the stigma and improve the well-being of people high in SPS. As 20–30% of the population is high in SPS [3,4], creating awareness about SPS could have a large impact on society. When people are able to recognise SPS in themselves or in others, they might be able to take the specific characteristics, as described in this paper, into account and capitalise on their positive potential. Indeed, from the perspective of a person-environment fit, this is important. Person-environment fit can be seen as the degree of congruence between a person and environment [40]. As known from the literature on person-environment-fit, outcomes, for example, in a work environment, are best when personal characteristics (such as SPS and the specific needs coming with that) and environmental characteristics (such as values) are compatible. In addition to creating awareness in society, more knowledge could be implemented in educational curricula for teachers and health care specialists, such as psychologists: they are the future professionals who are going to come across pupils or clients high in SPS. Specifically, when teachers are able to recognise SPS in their pupils, they can take their specific characteristics into account, for example, by creating a comfortable place for them in the classroom (not in the centre with many children and noise around them). Health care specialists could benefit from scientific insights into SPS in their counselling practice since individuals high in SPS are more prone to stress-related disorders but are also shown to respond more to treatments [10]. Furthermore, knowledge of SPS within work environments may help select the right people for the right jobs (optimal fit). Some characteristics of people high in SPS, such as empathy and attention to detail, can be beneficial in many work environments. A relevant hypothesis derived from differential susceptibility is that employees high in SPS may be more susceptible to work stress but also benefit disproportionally from training at work.

In conclusion, scientific insights into SPS can help understand SPS better, which might improve the person-environment fit for people high in SPS in all kinds of environments such as education and work. However, to allow scientific insights to reach society easily, more publicly available knowledge must be spread. This could be achieved, for example, by raising awareness in practical articles in magazines, social media, and presentations in teacher and health care education.

## 5. Conclusions

This study provided a rich understanding of how adults high in SPS experience the positive and negative characteristics of the trait. This conceptualisation can be further tested in quantitative research and contributed to generating ecologically valid hypotheses about phenotyping of the SPS trait. This study broadens the horizon on the theoretical perspective on SPS as well as a more practical view on coping with the negative sides of SPS and the well-being of individuals high in SPS.

## Figures and Tables

**Figure 1 jcm-10-04912-f001:**
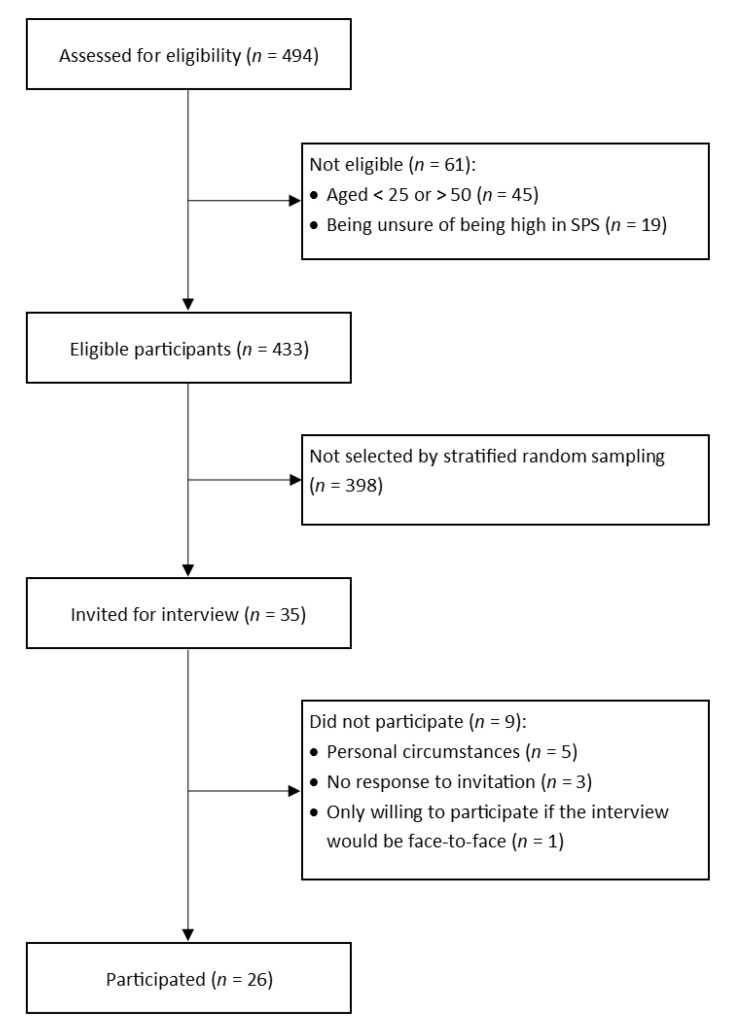
Flowchart of participant recruitment.

**Figure 2 jcm-10-04912-f002:**
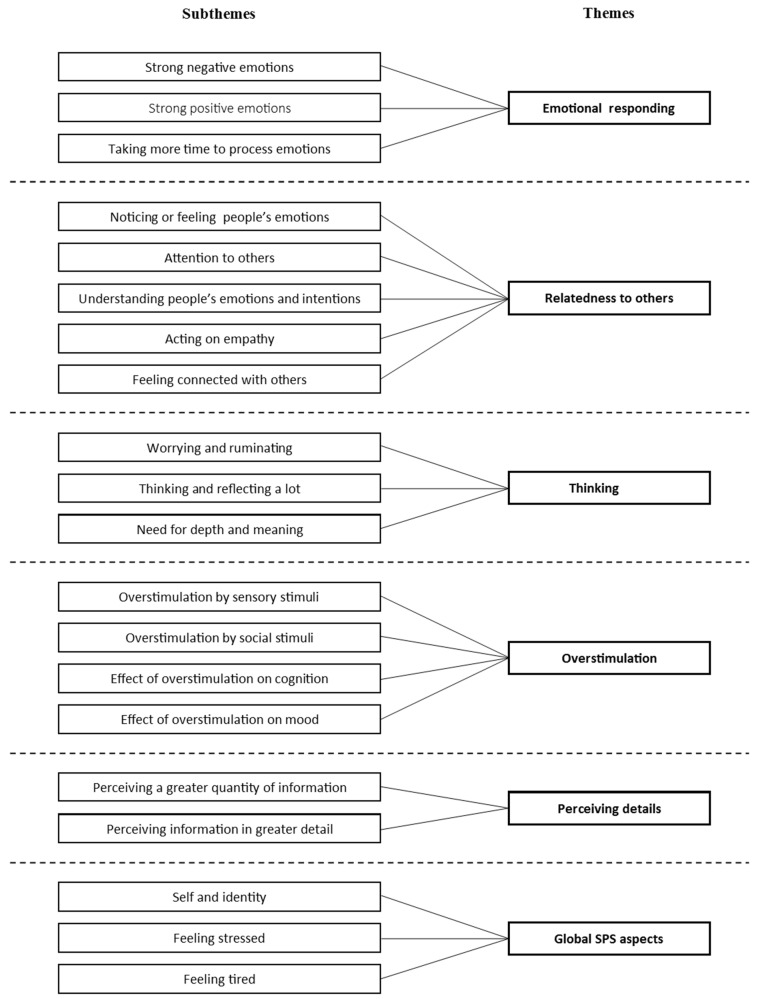
Identified themes and subthemes in SPS characteristics.

**Table 1 jcm-10-04912-t001:** Characteristics of the sample: SPS and Big 5 personality.

SPS and Personality	M (SD)
SPS (7-point scale)	
Highly Sensitive Person Scale	5.41 (0.49)
Sensory Processing Sensitivity Questionnaire	5.66 (0.55)
Personality (5-point scale)	
Extraversion	3.29 (0.59)
Agreeableness	4.13 (0.39)
Conscientiousness	3.65 (0.47)
Negative emotionality	3.14 (0.68)
Openness to experience	4.09 (0.47)

## Data Availability

Anonymised data can be obtained by contacting the corresponding or last author provided the applicant will submit a research plan and a researcher of the current study is involved in the research of shared data.

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
