# Peer review of "Experiences of Adults High in the Personality Trait Sensory Processing Sensitivity: A Qualitative Study"

_jcm, 2021, doi:10.3390/jcm10214912_

Round 1

Reviewer 1 Report

This paper is about a relevant research, on a relevant and “new” topic, and it gives a good contribution to the field. It is well structured and well written. Nevertheless, some issues should be improved.

P.2, line 65: “However, interpretation of questionnaire results is always limited by the content validity of the scale”

This statement questions the content validity of the existing scale (HSP), without explaining what this means, concretely: in the next paragraph, from line 75 on, some of the critics are presented, mas they should come above, on line 65, and clearly explain why this scale may be outdated and which studies lead to that conclusion.

P.3, line 116: The participants have age 25-50 and the authors do not explain why older people was not included. Moreover, on p.14, line 488, the authors state “the main strength of the current study is the demographically diverse sample”. This is incoherent and should be addressed.

P.4, line 123: “The Medical Ethics Review Committee of Radboudumc confirmed that the Medical Research Involving Human Subjects Act does not apply to this study, and thus no official approval of the committee was required (2019.6021.

Of course this is not medical research, this is psychological research; is there not a broader Ethic Committee to which the research could be submitted? And a statement like the following should be added: the privacy was observed in accordance with the principles of the Declaration of Helsinki.

P.4, line 127: Questionnaires:

The questionnaires are not well presented, there is lack of information. The authors had already made some statements about the HSP scale in the Introduction, but the information is needed on the Instruments section. If it has three dimensions, the internal consistency of each dimension (and not only the total one) should be provided, and some item examples. The authors mention the”new 30-item Sensory Processing Sensitivity Questionnaire (Cronbach’s α = 0.88)”. They do not give any information about this questionnaire, its construction process, why they have the need to use it and what is its incremental validity. This has to be properly explained.

  1. 6., line 200: “Comparing the scores on the Big 5 201 dimensions with a representative Dutch sample [26], our sample appeared to score higher on openness to experience and negative emotionality, which corresponds with a recent meta-analysis on personality correlates of SPS [5].”

What about the other three dimensions? More information should be presented.

P.12, Summary: “Six themes emerged, most of which matched with the hypothesised key characteristics”

Where are the hypotesized characteristics presented? They should be clearly presented after the goals of the study, in the Introduction.

p.13- line 459: “Based on the insights from our study, the SPS phenotype cannot be described by a combination of high negative emotionality and high openness to experience alone... Our study suggests that SPS encompasses a specific profile, supporting previous work on personality profiles of SPS”.

I could not understand these statements. On p.6, line 200, about the Results, the authors had: “Comparing the scores on the Big 5 dimensions with a representative Dutch sample [26], our sample appeared to score higher on openness to experience and negative emotionality…” And on the Discussion they state that the SPS phenotype cannot be described by a combination of high negative emotionality and high openness to experience alone? What is exactly the personality profile of SPS they found. This seems contradictory, and in fact the authors do not give us enough information about Big 5 in their sample.

  1. 14, line 507: “Another potential limitation is that experiences of individuals high in SPS were studied using self-report.”

I think it does not make sense to consider self-report as a limitation in a qualitative study about personal experiences…; what the authors could mean is that it would be good to complement the self-report with others report, and they already have this as a future suggestion.

  1. 14, line 510: For example, many participants spontaneously brought in other-report, comparing their experiences to those of the people around them”.

Other-report pertains to the report of significant people about the one being assessed/giving information, to complement/compare with the self report data…; so, comparing their own experiences with others´ experiences is not other-report. This sentence should be clarified.

Writing flaws:

  1. 1, line 43: sensitivity to environments was exclusively regarded a risk factor, should be “regarded as a risk factor”

P.2, line 90: “However, the qualitative interviews which the HSP scale…” should be “interviews on which”

Finally, in the Abstract, line 19: “in understanding SPS therefore lies potential for innovating the ways we use to promote mental health.”.

I think the authors should take this statement and present some specific ideas about the implication of their results to mental health professionals dealing with this persons. This contribute is lacking.

Author Response

Thank you for your helpful suggestions. Please see the attachment.

Reviewer 2 Report

The rationale for using a qualitative approach should be imrpoved and extended, as well as, the explanation of the process of analysis of qualitative data. 

The novelty and relevance of the discussed ideas should also be improved. Which could be the clinical implications of the study's findings? 

Author Response

(The authors gave the same response as above.)
